# Genome-Wide Identification of a Maize Chitinase Gene Family and the Induction of Its Expression by *Fusarium verticillioides* (Sacc.) Nirenberg (1976) Infection

**DOI:** 10.3390/genes15081087

**Published:** 2024-08-17

**Authors:** Jesús Eduardo Cazares-Álvarez, Paúl Alán Báez-Astorga, Analilia Arroyo-Becerra, Ignacio Eduardo Maldonado-Mendoza

**Affiliations:** 1Departamento de Biotecnología Agrícola, Centro Interdisciplinario de Investigación para el Desarrollo Integral Regional (CIIDIR), Unidad Sinaloa, Instituto Politécnico Nacional, Guasave 81049, Sinaloa, Mexico; eduardo.cazaresalvarez@gmail.com; 2CONAHCYT—Departamento de Biotecnología Agrícola, Centro Interdisciplinario de Investigación para el Desarrollo Integral Regional (CIIDIR), Unidad Sinaloa, Instituto Politécnico Nacional, Guasave 81049, Sinaloa, Mexico; a_lan6@hotmail.com; 3Laboratorio de Genómica Funcional y Biotecnología de Plantas, Centro de Investigación en Biotecnología Aplicada, Instituto Politécnico Nacional, Ex-Hacienda San Juan Molino Carretera Estatal Km 1.5, Santa Inés-Tecuexcomac-Tepetitla 90700, Tlaxcala, Mexico; alarroyo@ipn.mx

**Keywords:** maize, chitinase, glycosyl, hydrolase, stress, fungi

## Abstract

Maize chitinases are involved in chitin hydrolysis. Chitinases are distributed across various organisms including animals, plants, and fungi and are grouped into different glycosyl hydrolase families and classes, depending on protein structure. However, many chitinase functions and their interactions with other plant proteins remain unknown. The economic importance of maize (*Zea mays* L.) makes it relevant for studying the function of plant chitinases and their biological roles. This work aims to identify chitinase genes in the maize genome to study their gene structure, family/class classification, *cis*-related elements, and gene expression under biotic stress, such as *Fusarium verticillioides* infection. Thirty-nine chitinase genes were identified and found to be distributed in three glycosyl hydrolase (GH) families (18, 19 and 20). Likewise, the conserved domains and motifs were identified in each GH family member. The identified *cis*-regulatory elements are involved in plant development, hormone response, defense, and abiotic stress response. Chitinase protein-interaction network analysis predicted that they interact mainly with cell wall proteins. qRT-PCR analysis confirmed in silico data showing that ten different maize chitinase genes are induced in the presence of *F. verticillioides*, and that they could have several roles in pathogen infection depending on chitinase structure and cell wall localization.

## 1. Introduction

Chitin is the second most abundant polysaccharide in nature and can be found in crustacean exoskeletons and fungal cell walls [1], in insects and animals [2]. A linear polymer of β-1,4-N-acetylglucosamine (GlcNAC), chitin occurs in three forms: α, β and γ. The most abundant form, α-chitin, can be found in the exoskeletons of insects and crustaceans; this form exhibits adjacent sheets along the c axis (crystallographic orientation) that are arranged in the same direction [3]. β-chitin has adjacent sheets along the c axis in the same direction, resulting in the parallel orientation of sheets [4]. β-chitin is found in squid pens [5] and insects [6]. γ-chitin has three chitin chains with alternating parallel and antiparallel aligned polymer chains, making this type of chitin the most complicated in structure [7].

Chitinases (EC 3.2.1.14) hydrolyze chitin glycosidic bonds, producing low-molecular-weight compounds. Chitinase genes are required for growth and development in chitin-synthesizing species. Chitinases are classified according to their amino acid sequences and belong to the GH18 (glycosyl hydrolase-18), GH19 and GH20 families, according to previously proposed criteria [8] as well as the CAZy database. Chitinases can be grouped into several classes: those that are within the GH19 family belong to classes I, II, IV, VI and VII, while GH18 members belong to classes III and V. The GH20 family consists of β-N-acetyl-D-hexosaminidases, which hydrolyze both GlcNAC and GalNAC [9].

Chitinases can also be classified according to the presence of specific modules (motifs) or conserved domains such as a pre-sequence, a catalytic domain, a linker domain, or the presence of a carbohydrate-binding domain [10]. Plant chitinases play an important role in plant defense against fungi containing chitin in their cell walls [11]. Chitinases belonging to classes I and IV possess a chitin-binding domain. In plants, this domain plays an important role in generating the chitin-derived signaling molecules (chito-oligosaccharides) that are typically released through binding of these enzymes to chitin in fungal cell walls, which is subsequently followed by chitin hydrolysis [12]. Chito-oligosaccharides (CHOs) are one of the main elicitors in plants that induce a series of defense mechanisms leading to the production of pathogenesis-related proteins, reactive oxygen species (ROS) and phytohormones [13,14,15].

Maize is an economically important crop worldwide, and it grows in a range of environments in many countries including Mexico, from where it originated. Maize is mainly used as human food, livestock feed, and in the production of bioethanol for energy [16]. This crop is susceptible to various fungal diseases, including *Fusarium* stalk, ear and root rot (SERR), a disease caused by *F. verticillioides* [17]. Different species of *Fusarium* (*F. verticillioides*, *F. nygamai* and *F. thapsinum*) are reported to coexist in mixed infections in a single maize plant [18]. Phytopathogenic fungi can induce the production of plant antifungal compounds including enzymes such as chitinases. In maize, 27 putative genes encoding endochitinases, including members of the GH18 (Class III) and GH19 families (classes I, II and IV), are induced by *Trichoderma harzianum* [19], *Ustilago maydis* [20] and *Aspergillus flavus* [21] colonization.

*F. verticillioides* is an aggressive phytopathogen that has evolved a mechanism to modify maize chitinases (ChitA and ChitB from the GH19 family) through the production of an effector chitinase-modifying protein (Cmp) [22]. This modification cuts the chitin-binding domain in the N-terminus upstream of a conserved polyglycine-rich site. Therefore, maize chitinases lacking the chitin-binding domain, although still active, cannot bind to the chitin substrate in the fungal cell wall. Cmp production has been confirmed in other fungal phytopathogens such as *Bipolaris zeicola* [23], *Stenocarpella maydis* [24] and *Fusarium oxysporum* [25]. The effect of Cmp activity is to limit fungal cell-wall chitin degradation and the production of the chito-oligomer elicitors, resulting in the reduction in plant defenses to allow these pathogens to infect maize cells.

Few studies have performed a genome-wide analysis of maize chitinases [19,21], and this could be explained by the fact that we know very little about the role of chitinases and their involvement in biotic and abiotic stresses, including maize–pathogen interactions. This is especially true in root tissues, such as during *F. verticillioides* infection. The availability of maize transcriptome data now makes it feasible to assess gene expression under different stress conditions. In the present study, we performed a genome-wide analysis to identify maize chitinase genes in order to characterize them according to their classification in GH families and classes. We examined the *cis*-elements present in their upstream promoter sequences, the presence of conserved motifs and domains in their encoded proteins, and their possible interaction with other maize proteins using co-expression network analysis. We also used transcriptome data to understand the putative involvement in the *F. verticillioides*-maize root interaction. Finally, this work provides insight and contributes to our comprehension of the genome-wide regulation of maize chitinase expression.

## 2. Materials and Methods

### 2.1. Identification of Putative Chitinase Gene

Nucleotide sequences of maize chitinase genes and their corresponding amino acid sequences were obtained from the Maize Genetics and Genomics Database (MaizeGDB) (https://www.maizegdb.org, accessed on 24 May 2023), using the B73 genome and “chitinase, chitin, endochitinase, and exochitinase” as discovery terms [26]. A total of 202 gene models were retrieved. From these, we removed all genes that were not annotated inside the Zm-B73-REFERENCE-NAM-5.0 genome database, resulting in 39 gene models. Thirty-nine amino acid sequences were analyzed using the Pfam database (https://www.ebi.ac.uk/interpro/) to confirm the presence of the glycoside hydrolase domain. Chromosomic localization, exon number, and transcript length were obtained from the Ensembl Plants database (https://www.plants.ensembl.org/). The molecular weight (kDa) and isoelectric point of chitinase amino acid sequences were obtained using the “Compute pI/MW” tool from the ExPAsy database (https://www.expasy.org/tools). The physical locations of chitinase genes were schematically drawn on their respective maize chromosomes using the online server MG2C (https://mg2c.iask.in/mg2c_v2.0/).

### 2.2. Gene Structure, Multiple Alignment, and Phylogenetic Analysis

Gene structure (exon–intron) was analyzed based on the coding sequence (CDS) of the chitinases at the Gene Structure Display Server (GSDS, https://gsds.gao-lab.org/) [27]. Evolutionary relationships among maize chitinase families were analyzed by performing an unrooted Neighbor-Joining phylogenetic tree constructed with the MEGA 11.0 software based on the Jones–Taylor–Thornton (JTT) model, using the MUSCLE alignment function with 1000 bootstrap replicates and default parameters for each chitinase family. The chitinase family tree was constructed for each chitinase class using orthologous genes from *Sorghum bicolor*, *Oryza sativa* and *Triticum aestivum*.

### 2.3. Prediction of cis-Acting Elements, Conserved Domains and Transcription Factor Binding Sites

To predict the *cis*-acting regulatory elements in the promoter region of the 39 *Z. mays* chitinase genes, a 2.0-kb sequence upstream of the translation initiation site (ATG) was analyzed, along with the 5′UTR region, using the PlantCARE website (http://bioinformatics.psb.ugent.be/webtools/plantcare/html) [28]. All amino acid sequences were analyzed using the NCBI-CDD database (https://www.ncbi.nlm.nih.gov/Structure/cdd/cdd.shtml) to find conserved domains, and the MEME website (https://meme-suite.org/index.html) to find conserved motifs [29]. The domains illustration was made using the Domain Illustrator software 2.0 version [30]. Subcellular localization prediction of each chitinase protein was obtained using WoLF PSORT (https://wolfpsort.hgc.jp/) [31], Plant-mSubP (https://bioinfo.usu.edu/Plant-mSubP/) [32] and MULocDeep (https://www.mu-loc.org/) [33], and signal peptide prediction was made by the SignalP 6.0 tool (https://services.healthtech.dtu.dk/services/SignalP-6.0/) [34]. To identify the binding sites of transcription factors in the promoter region of chitinases (2.0 kb upstream), the PlantRegMap database was used (http://plantregmap.gao-lab.org/) [35] with the following parameters: *p*-value ≤ 1 × 10^−4^ and a maximum number of 15.

### 2.4. In Silico Expression Analysis of Chitinase Genes Using RNA-Seq Data

In silico gene transcription analysis of the maize glycoside hydrolase family was performed using RNA-seq data in order to retrieve the expression data in response to abiotic stress. The fragment per kilobase of exon per million fragments mapped (FPKM) was obtained from the B73 V5 genome through a gene ID (*Zm00001*) search using qTeller in the MaizeGDB database [36]. Previous reports were used to study gene expression for abiotic stress using seedling [37] and leaf [38] samples. The extracted data were then log^2^ transformed and used to generate heatmaps via the TBtools package [39].

### 2.5. Quantitative Real-Time PCR Analysis

Previously, RNA-seq data were generated by our group from RNA samples of a rolled paper-towel assay using roots of 7-day-old maize seedlings (Garañón hybrid from Asgrow) inoculated with *F. verticillioides* P03 [40]. Differentially expressed genes (DEGs) were defined as those genes with adjusted *p*-value < 0.01 and Log2 fold change ±1. The differentially expressed genes (DEGs) obtained in the RNA-seq were analyzed and a list of chitinase DEGs with gene expression presented as fold-change value was generated. To confirm the induction of the maize root chitinase genes that respond to the *Fv* infection in the RNA-seq, gene expression analysis of the chitinase DEG genes was performed by RT-qPCR, using the SYBR Green Master Mix (Qiagen, Cat. No. 204074, Hilden, Germany) in a Rotor Gene-Q real-time PCR system instrument (Qiagen, Cat. No. 9001550, Hilden, Germany). The RNA used for RT-qPCR was the same as that used for RNA-seq. The RT-qPCR reactions were performed using two biological replicates with three technical replicates, and were run using 5 μL of SYBR Green Master Mix (2×), 10 μM of each primer and 10 ng of cDNA in a 10 μL final volume. The PCR program included a pre-heating step at 95 °C (5 min), followed by 40 cycles of 95 °C (30 s), 60 °C (30 s) and 72 °C (30 s). Relative quantification of chitinase genes was normalized to the cyclin-dependent kinase (CDK) gene [41], and the comparative threshold cycle method 2^−∆∆Ct^ [42] was used to calculate the fold-change (FC) values in gene expression. The set of primers used in this work are listed in Appendix A.

### 2.6. Protein–Protein Interaction Network

To analyze protein–protein interactions, the STRING database (http://string-db.org) [43] was used to generate a co-expression network. All maize chitinase proteins were submitted to the STRING database. The minimum required score was set to medium confidence (0.400). The maximum number of interactors showed no more than 20 on the first shell, and no more than 10 on the second shell. The protein–protein network was obtained using text mining and experiments, as well as database and co-expression interaction sources.

## 3. Results

### 3.1. Genome-Wide Identification of Chitinase Genes in Z. mays

A total of 39 chitinase genes were obtained from the Zm-B73-REFERENCE-NAM-5.0 genome. Gene name and ID, chromosomal position, exon and transcript count, glycoside hydrolase family, molecular weight, and isoelectric point are presented in Table 1. The amino acid sequence length of the maize chitinase genes ranged from 96 (*Zm00001eb270440*) to 599 (*Zm00001eb008880*) amino acids. We found 17 chitinase genes belonging to the GH18 family (PF00704), 18 genes belonging to the GH19 family (PF00182), and 4 genes belonging to the GH20 (PF00728) family. Their predicted molecular weight (MW) ranged from 10.55 kDa (*Zm00001eb270440*) to 66.12 kDa (*Zm00001eb008880*), and their predicted isoelectric points ranged from 4.06 (*Zm00001eb157820*) to 10.35 (*Zm00001eb272090*). Most GH19 chitinases presented a basic isoelectric point, specifically *Zm00001eb078730* and *Zm00001eb425600* (Cta1 and Ctb1) from class IV, as well as nine GH18 chitinases and one from the GH20 family. An acidic isoelectric point was exhibited by GH19 family members of classes I, II, and IV; GH18 family members of class III; and three GH20 family members.

### 3.2. Phylogenetic Analysis and Multiple Sequence Alignment

An unrooted phylogenic tree was constructed to study the evolutionary relationships among the chitinase families in the maize genome. The chitinase protein phylogenetic tree was constructed using proteins from *S. bicolor* (36 sequences), *T. aestivum* (37 sequences) and *O. sativa* (32 sequences) (see Figure 1). The class classification revealed that chitinase members of the GH18 family belong to class III (containing hevamine, narbonin-like and SI-CLP protein sequences) and class V, and members of the GH19 family belong to classes I, II and IV. Although the *Zm00001eb317090* gene belongs to family 19 and is grouped in the class IV clade, it does not contain a chitin-binding domain, and shares more protein characteristics that make it closer to class II proteins.

### 3.3. Chromosomal Location and Intron–Exon Architecture

All 39 chitinase genes were assigned to nine out of ten maize chromosomes (Figure 2). Chromosome number 6 contained the highest number of chitinase genes (seven), followed by chromosomes 3 and 8 (six chitinase genes in each chromosome). No chitinase genes were found in chromosome 9.

To study the diversity of maize chitinase genes, we next analyzed the exon–intron sequences. Among the thirty-nine chitinase genes in *Z. mays*, only four genes had more than two introns; four genes had two introns; eleven genes had one intron; and twenty genes did not have any introns. *Zm000011eb266300* was the longest gene, containing 15 introns, followed by *Zm000011eb365840* and *Zm000011eb288150*, with 14 and 13 introns, respectively, and *Zm000011eb358410*, with 7 introns (Figure 3).

Alternative splicing can occur in maize chitinase genes due to the presence of introns; *Zm00001eb358410* (four splice variants), *Zm00001eb002620* (three splice variants), *Zm00001eb266300* (three splice variants), and *Zm00001eb365840* (six splice variants) all presented more than one splicing transcript (Appendix A). Since most chitinase genes had only one identified transcript, only one transcript (T001 from Appendix A) from each gene was used to analyze each glycosyl hydrolase family.

### 3.4. Conserved Domains and Motifs

In order to better understand the structural differences between members in the chitinase gene families, we next identified the conserved domains and motifs (Figure 4). Motifs 6, 7, 8, 10 and 17 were exclusive to the GH18 gene family members, whereas motifs 1, 2, 3, 4, 5, 9, 13 and 16 were mainly found in the GH19 gene family. Members of the GH20 gene family contained conserved motifs 12, 14, 15, 18, 19 and 20 (Figure 4A, Appendix A). As shown in Figure 4B, all protein sequences possessed domains typical of the glycosyl hydrolase families. According to their enzymatic activity, three different domains were present in members of class III in the GH18 gene family: hevamine, narbonin, and SI-CLP; other GH18 members had a domain typical of class V. However, GH18 family members did not contain any chitin-binding domain, which was only found in GH19 family members (ten out of eighteen). All GH20 family members presented the glycosyl hydrolase-20 and glycosyl hexaminidase HexB-like domains.

Additionally, almost all GH19 chitinase genes that presented chitin-binding domains were predicted to be extracellular proteins. However, Zm00001eb272050 did not contain a predicted signal peptide, but it had a chitin-binding domain and is predicted to be a cytoplasmic protein, and Zm00001eb078720 contained one signal peptide, and is predicted to be chloroplastic (Appendix A).

### 3.5. Cis-Regulatory Elements and Transcription Factors in the Promoter Region

*Cis*-regulatory element analysis was performed to elucidate the possible regulatory mechanisms/pathways/signals of maize chitinase genes when elicited in response to hormones, abiotic stress, pathogen infections and plant development (Appendix A). The hormone and abiotic stress *cis*-acting regulatory elements were widely present in chitinase genes. Defense-related elements had fewer repetitions among genes, and only a W-box element (involved in WRKY transcription factor recognition) was repeatedly found. The hormone-related elements ABRE (abscisic acid response), CGTCA-motif (methyl-jasmonate response), MYC (jasmonate signaling), as-1 and TGACG-motif (response to pathogens and methyl-jasmonate response) were the most abundant. We also found the abiotic stress *cis*-elements ARE (anaerobic induction), G-box (response to light) and STRE (general stress) in abundance. The AAGAA-motif (secondary xylem development) and CCGTCC-box (meristem activation) were the most frequently detected *cis*-regulatory elements related to plant developmental processes in chitinase upstream sequences. The methyl jasmonate response *cis*-elements were present in most chitinase promoters, followed by abiotic-stress and light-response elements. We also analyzed their gene expression (Appendix A) and found that some chitinase genes may be transcriptionally induced under different stress conditions (cold, heat, salinity, UV light, and drought) (Appendix A). The TF binding-site prediction revealed a total of 6281 binding sites for 18 TFs (Appendix A) including AP2/ERF [44,45], B3 [46,47], BBR-BPC [48], bHLH [49], bZIP [50,51,52], C2H2 [53,54,55], Dof [56,57,58], G2-like [59,60,61], GATA [62], HD-ZIP [63,64,65], LBD [66,67], MIKC_MADS [68,69], MYB and MYB-related [70,71], NAC [72,73], TALE [74,75], TCP [76,77] and WRKY [78,79,80].

### 3.6. In Silico Analysis of Maize Chitinase Gene Expression in 7-Day-Old F. verticillioide Infected Roots

RNA-seq data from maize roots infected for 7 days with *F. verticillioides* (*Fv*) [40] were used to determine chitinase gene-expression patterns in maize roots (Appendix A). Overall, RNA-seq data showed that 15 out of 39 chitinase genes were differentially expressed in maize roots in response to seven-day infection with *Fv*. All fifteen chitinase genes were upregulated. During RT-qPCR analysis, it was only possible to amplify 14 out of 15 differentially expressed chitinase genes, due to the sequence similarity among chitinase families. We were not able to amplify the *Zm00001eb272050* gene by PCR, and therefore this gene was not included in the RT-qPCR analysis. Table 2 shows the RT-qPCR expression analysis of the 14 chitinase genes from maize roots identified in RNA-seq data under the *Fv* infection condition.

Ten out of the fourteen chitinase genes analyzed by RT-qPCR showed the same induced expression trend as observed in the RNA-seq data (Table 2). Three out of six GH18 gene family members, six out of seven GH19 members, and one GH20 member were induced by seven days of *Fv* infection.

### 3.7. Protein–Protein Interaction Network Analysis

Protein–protein interaction networks with the ten RT-qPCR validated chitinase genes from the maize–*Fv* interaction at seven days’ infection predicted 20 functional maize protein partners, exhibiting 100 interactions (Figure 5; Appendix A). *Zm00001eb317090* showed 30 interactions, followed by *Zm00001eb008880* with 13 interactions and *Zm00001eb167340* with 10 interactions. GH18 chitinases (*Zm00001eb167340* and *Zm00001eb168350*) were predicted to interact with β-hexosaminidases, while GH19 (*Zm00001eb272090*) was the main maize chitinase partner to interact with cellulose synthase proteins, and GH20 (*Zm00001eb008880*) was related to β-galactosidase proteins (Appendix A). These results show diverse patterns of maize chitinase co-expression with maize cell wall proteins.

## 4. Discussion

In plants, glycosyl hydrolases are present as large gene families that are involved in several biological and defense mechanisms, and they can be expressed in different cell compartments [81]. The GH18 and GH19 chitinase families are some of the most studied plant enzymes, due to their involvement in different metabolic processes, in addition to their synergistic function in plant development and defense responses [82]. GH18 chitinases have been reported in fungi [83], bacteria [84] and plants [85]. GH19 chitinases are predominantly found in plants (e.g., class I chitinases are found in *Camellia sinensis* [86], class II in soybean [87] and class IV in grape [88]), while GH18 class III chitinases have been reported in *Rhododendron irroratum* and *Pteris ryukyuensis* [89,90]. On the other hand, GH20 chitinase family members are β-hexosaminidases; these enzymes have been detected before maize seed germination and in scutellum in seedlings [91]. β-Hexosaminidases are proposed to participate in N-glycan processing, depending on their subcellular localization: N-glycan trimming occurs in the vacuole for vacuolar proteins, whereas N-glycans are processed on secretory glycoproteins for plasma membrane proteins [92].

Several maize chitinase genes described here have more than two introns, and some of these genes also presented alternative splicing (Appendix A). Alternative splicing occurs when there is a deviation from constitutive splicing, in which the removal of certain introns is omitted, producing various forms of mature mRNA [93]. This process is involved in mediating diverse biological processes [94] and has an important role in plant abiotic-stress responses [95], as well as a direct effect on protein function, amino acid sequence, and enzymatic properties [96].

Protein domains are short amino acid sequences that can be considered as a functional site, or as structural and evolutionary units [97]. These units are considered important for protein function [98], and proteins can acquire several domains over time through gene fusion and exon recombination [99]. However, protein domains can suffer changes like mutations or deletions, and cause dramatic effects on gene function [100]. Plant chitinases usually have different domains such as catalytic, chitin-binding, and linker domains [10]. In the present work, all maize chitinases had at least one domain belonging to a glycosyl hydrolase family. GH18 sequences presented only one type of GH-conserved domain per gene, and no chitin-binding-related domains. Gene members of the GH20 family possess two domains: the glycosyl hydrolase-20 and the glycosyl hexaminidase HexB-like domain. Most GH19 chitinases presented either a glycosyl hydrolase-19 domain and/or a chitin-binding domain; the latter is able to bind and hydrolyze insoluble or soluble chitin forms, such as the GH19 chitinase from *Hevea brasiliensis* [101] or *Streptomyces griseus*. The chitin-binding domain of *S. griseus* differs slightly from those of plants, and it can also bind to cellulose [102]. For example, inserting the chitin-binding domain from *Serratia marcescens* into the structure of a chimeric chitinase from *Trichoderma atroviride* (Chit42) resulted in increased enzymatic activity and colloidal chitin binding, as well as higher antifungal activity against phytopathogenic fungi [103]. This demonstrates the importance of the chitin-binding domain during chitin degradation.

In plants, the chitin-binding domain is important due to its relatedness to mechanisms of avoidance of fungal phytopathogen infection, such as in the interactions between *Fusarium solani* and *Nicotiana tabacum* [104], *Candida albicans* and *Moringa oleifera* [105], and *Trichoderma viride* and *Brassica juncea* [106]. These studies suggest that chitinases with a chitin-binding domain could potentially be used as antifungal enzymes, since they can bind to chitin from the fungal cell wall and hydrolyze it to produce small chito-oligomers that elicit the plant defense [107]. These may occur in GH19 class I and IV chitinases, due to the presence of a chitin-binding domain [108].

Motifs are short amino acid sequences shared by protein family members that have a specific structural function [109]. In our study, maize chitinases presented the NYNG conserved motif (Motif 1, Appendix A). This motif is generally located within a loop III structure close to the amino acid catalytic triad of the GH19 family (Glu221, Arg361 and Glu234) [110] that belongs to the lysozyme superfamily. GH19 maize family members, like most of the GH22, 23, 24 and 46 families, have two other main elements: an α-helix and a β-hairpin motif [111], which are important, since they contain glutamate residues acting as general acid catalysts that allow proteins to perform chitin catalysis. Thirty-two chitinases have signal peptides that localize them for their secretion. Likewise, the hevamine-type chitinase from *H. brasiliensis* also contains a signal peptide sequence at the N-terminus [112] for its vacuolar secretion. In maize, according to the WoLF PSORT (an extension of the PSORTII program) *in silico* subcellular prediction software, one chitinase is targeted to the vacuole (Zm00001eb167340), nine are targeted to the chloroplast, and eighteen are extracellular proteins (Appendix A). GH18 proteins are predicted to have a different subcellular distribution as compared to GH19 proteins, where most chitinases are predicted to be extracellular. Plant chitinases are proposed to be targeted to the chloroplast, endoplasmic reticulum and vacuole, and these subcellular localizations have been tested in *Arabidopsis thaliana* [113].

Different *cis*-regulatory elements in maize chitinases have been found to be related to plant defense, development, response to abiotic stress, and hormones. The plant development elements AAGA-, CAT-box, CCGTCC, RY-element, HD-Zip1 and GCN-4 motifs were present in maize chitinase genes. The GCN-4 motif is fundamental to specific meristem gene expression [114]. Hormone-related elements such as ABRE, as-1, MYC, CGTCA-, and TGACG-motifs were also present in most chitinase genes. These *cis*-elements, such as ABRE (abscisic acid-response-related element), are related to several dehydration-response genes that trigger ABA production [115]. Likewise, Chi2 and Chi14 induction in *Cucumis sativus* L. was also observed in response to jasmonic acid, ethylene and salicylic acid treatments [116], suggesting that maize chitinases could be involved in hormone signaling in response to either abiotic or biotic stresses during development. ARE, G-box, STRE, Box4, and DRE-core were the most abundant *cis*-elements in abiotic stress-related categories such as general stress, as well as light and dehydration responses. These elements have also been reported in chitinase genes from *Allium sativum* L. [117], *Thalassiosira pseudonana* [26] and *B. juncea* [118]. The W-box in chitinase genes plays a role in the induction of plant defense genes against pathogenic fungi [119]. TC-rich repeat elements are also involved in the induction of defense genes, and they have been reported in *C. sativus* L. [120]. The GH18 and GH19 family members are induced under osmotic stress in *Ammopiptanthus nanus* [121] and *Solanum lycopersicum* [122]. According to our in silico maize chitinase expression analysis (Appendix A), most GH18 genes are induced under cold, heat, UV, and salt stresses, such as a class III gene from *A. thaliana* that is induced in response to different environmental stress conditions [123]. Interestingly, most GH19 and GH20 genes are either repressed or do not show induction under different types of stress (drought and cold). This may be related to the activity of GH19 and GH20; GH19 genes are mainly induced in the presence of phytopathogenic organisms [124,125], while GH20 genes are involved in N-glycan processing [92].

Transcription factors (TFs) can be induced under abiotic or biotic stresses. TFs can be regulated by phosphorylation, inhibitory factors or de novo synthesis [126]. Maize chitinase genes exhibit target sites for several transcription factor families. These TFs are related to hormone response, plant development, and defense, including the ERE and WRKY TFs that are involved in the plant-pathogen defense response [127,128].

Previous studies have shown the induction of chitinases in maize plants infected with *F. verticillioides* and *A. flavus* [129,130], which increase the plant resistance to fungal pathogen infection. This effect may be related to the release of chito-oligosaccharides acting as pathogen-associated molecular patterns (PAMPs), which are derived from fungal cell wall degradation by chitinases [131]. Chitinases can accumulate in different plant organelles such as the chloroplast, vacuole and nucleus, or they can be secreted into the extracellular space [113]. The extracellular space, or apoplast, is considered to be a cellular compartment that plays important roles such as nutrient metabolism, plant growth, defense and signaling [132]. In the present study, we identified by RNA-seq 15 maize chitinase genes induced in response to a 7-day *F. verticillioides* infection; additionally, we confirmed by qRT-PCR the induction of ten genes. Five GH19 chitinase members (*Zm00001eb317090*, *Zm00001eb272090*, *Zm00001eb346860*, *Zm00001eb078730* and *Zm00001eb425600*) are predicted to be extracellular proteins. Four of these genes are indicated in Figure 6: (a) *Zm00001eb272090* [Chitinase 2], (b) *Zm00001eb346860* [Chitinase 21], (c) *Zm00001eb078730* [Chitinase A1], and (d) *Zm00001eb425600* [Chitinase B1]). These genes possess a chitin-binding domain and motif 5, which are important for substrate recognition and chitin binding. ChitA and ChitB were previously reported to be targeted in a polyglycine-rich region by *F. verticillioides* chitinase modifying enzymes (Cmps) (Figure 6) [23]. This suggests that (a) *Zm00001eb272090* and (b) *Zm00001eb346860* may also be targets of Cmps, since they also possess a potential polyglycine site targeted by this effector protein. Taken together, these results suggest that these four extracellular chitinases could be targets of *F. verticillioides* Cmps, preventing them from (1) hydrolyzing the chitin present in the fungal cell wall, and (2) inducing the presence of oligosaccharides (PAMPs) that trigger the plant defense mechanisms. As a result, the maize root does not perceive the invasion by the fungus, allowing for the *F. verticillioides* infection to proceed. *Zm00001eb317090* (Brittle stalk4; [g] in Figure 6) does not possess this Cmp region, and our co-expression network analysis predicted its interaction with other cell wall proteins (Appendix A) that may be involved in cell wall strengthening [133], including two cytoplasmic/membrane GH18 members (*Zm00001eb250900* [chitinase 28] and *Zm00001eb168350* [chitinase 29]) (e and f in Figure 6, respectively). 

The other induced chitinases may also participate in the response to pathogen attack, playing different roles in the vacuole/apoplast ([j] *Zm00001eb167340*, chitinase 27 [pI = 8.69]). In addition, other intracellularly localized induced chitinases ([i] *Zm00001eb354540* [chitinase 23] and [h] *Zm00001eb008880* [Exochitinase 2]; chloroplastic/extracellular) may play other functions in this particular interaction, such as BC15/OsCTL1 (chitinase-like 1), which is necessary for cellulose biosynthesis in rice [134]. Chloroplastic chitinases may also be regulated by the accumulation of ROS in response to fungus infection [135].

Co-expression network analysis showed interactions with other maize proteins according to text mining, experiments, databases, and co-expression. The most prominent interactions were between β-hexosaminidase and cellulose synthase, which play a major role in cell wall formation [136]. β-Hexosaminidase (HEXO) has been detected in different plant tissues and has a role during seed germination. HEXOs also show different types of catalytic activity and substrate affinity to N-glycans [137], and they are capable of hydrolyzing N-acetyl-hexosamine residues to produce low-molecular-weight compounds [138]. However, their physiological and biological functions are not fully understood, and it has been hypothesized that they are mainly involved in chitin degradation. On the other hand, cellulose synthase (generally found as an enzymatic complex) is responsible for cellulose synthesis at the plasma membrane [139]. *Zm00001eb317090* (GH19, Brittle stalk4 [g]) is related to *Zm00001eb250900* (GH18, Chitinase 28 [e]) and *Zm00001eb168350* (GH18, Chitinase 29 [f]), which are predicted to be cytoplasmic/membrane proteins of class III narbonin chitinases. In addition, *Zm00001eb317090* (g) possesses the GhCTL (chitinase-like proteins) consensus [140], which differs from the classical GH19 consensus; this difference in catalytic residues may allow this protein to participate in the synthesis of primary and secondary cell walls. However, although chitinases are known to bind to N-acetyl-D-glucosamine (GlcNAC) residues, plants do not produce chitin. Therefore, there must be a mechanism by which chitinases can be induced in plants, even if they are not under pathogen attack. For example, one previous study demonstrated that N-acetyl-D-glucosamine-containing arabinogalactan proteins are sensitive to endochitinases, and that they could be implicated in embryogenesis [141]. Since arabinogalactan proteins are typically found in membranes or cell walls, these may be some of the few proteins that contain GlcNAC, and they could explain why chitinases participate in cell wall biosynthesis

## 5. Conclusions

In summary, we analyzed maize chitinase genes in order to learn about their putative function and possible role in fungal infection. Our findings suggest that they may participate in different plant biological pathways, even though we addressed their involvement during fungal pathogen infection (e.g., *F. verticillioides*). We also found by qRT-PCR that maize chitinases can participate in *Fv* infection, according to their domain type and subcellular localization. This works opens new avenues to study novel biochemical mechanisms in which chitinases may play a major role.

## Figures and Tables

**Figure 1 genes-15-01087-f001:**
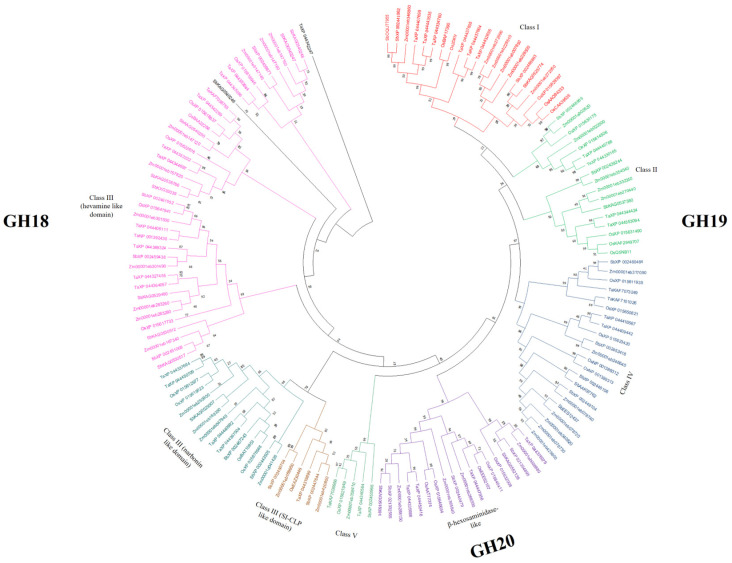
Phylogenetic tree of *Z. mays* chitinase genes. GH18 includes family members of class III (hevamine-like domain; narbonin-like domain; SI-CLP-like domain); GH19 includes family members of class I, class II and class IV; and GH20 includes family members that are β-hexosaminidase-like proteins. The phylogenetic tree was built using the neighbor-joining (NJ) method and is based on the Jones–Taylor–Thornton (JTT) model, using the MUSCLE alignment function. The numbers at the tree nodes represent bootstrap percentage values. Bootstrap values are from 1000 replicates. Orthologous genes are labeled as Ta: *T. aestivum*; Sb: *S. bicolor*; Os: *O. sativa*.

**Figure 2 genes-15-01087-f002:**
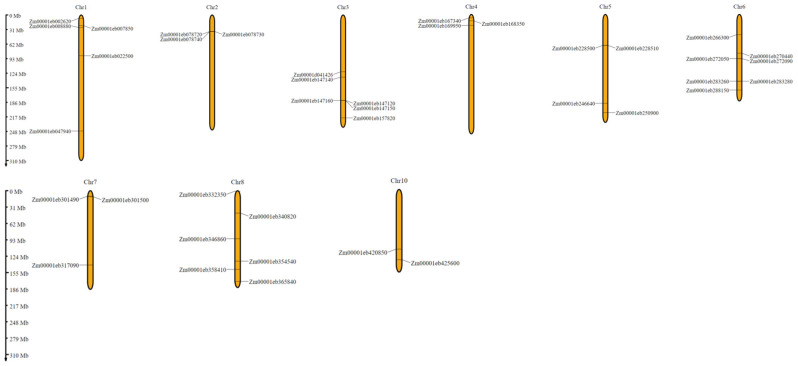
Chitinase gene distribution in maize chromosomes. Gene names are indicated on the left and right sides of the chromosome. The Y-axis represents the distance (Mb) between genes. Their locations may be close in each chromosome, but they are not contiguous with each other.

**Figure 3 genes-15-01087-f003:**
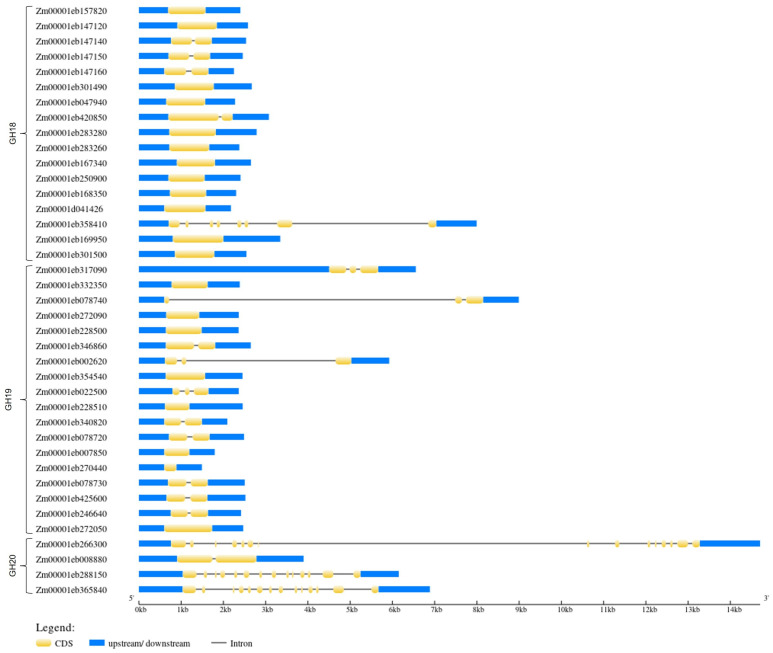
The intron–exon structure of the chitinase-coding gene families in *Z. mays*. The blue bars indicate the 5′ and 3′ UTR untranslated regions, the yellow bars indicate coding sequences (CDS), and the gray lines indicate intron sequences.

**Figure 4 genes-15-01087-f004:**
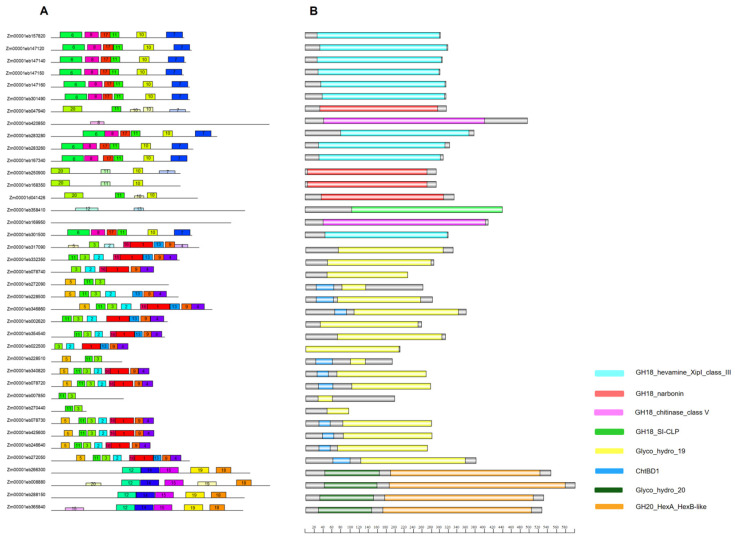
Schematic structure of conserved maize chitinase motifs and domains. (**A**) Motifs. Colored boxes represent conserved motifs and gray lines represent non-conserved sequences; motif number sequences are shown in Appendix A. (**B**) Domains. The conserved domains are represented by different colors.

**Figure 5 genes-15-01087-f005:**
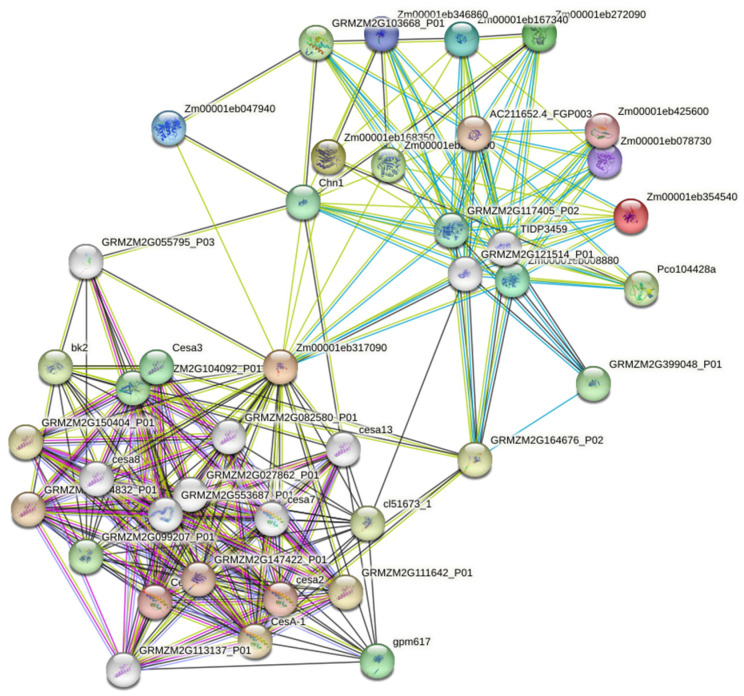
Functional protein-association-network analysis of maize chitinases using the STRING database. Yellow lines: text mining network; black lines: co-expression; blue lines: from curated databases; violet lines: experimentally determined. Colored nodes represent the first shell of proteins interactors, and the white nodes represent the second shell of interactors.

**Figure 6 genes-15-01087-f006:**
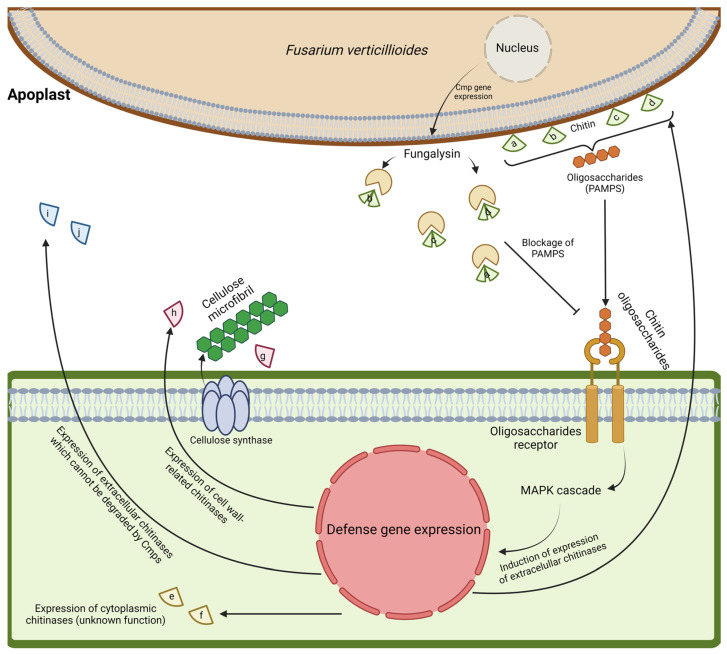
A proposed model of maize chitinase induction in the presence of *F. verticillioides*. Four chitinases (a–d) possess the chitin-binding domain and Cmp site (fungalysin); these proteins may act synergistically to hydrolyze chitin in the fungal cell wall to produce chitin oligosaccharides (elicitors). These enzymes may also be the target of fungalysin, and their binding capacity could be eliminated; however, maize may produce other chitinases (i and j) that do not possess the chitin-binding domain/Cmp site to hydrolyze chitin. Due to the absence of a carbohydrate-binding domain, the chitin-binding capacity is low in these two chitinases, but their catalytic activity remains intact, and the plant could be inducing these proteins as an alternative strategy to degrade fungal chitin. On the other hand, proteins g and h, and e and f, may be induced in a coordinated mechanism to interact with cellulose synthases and to promote cell wall biosynthesis in response to *Fv* infection, in order to reinforce the plant cell wall. Different letters represent different chitinase proteins. Letters a–d: *Zm00001eb272090*, *Zm00001eb346860*, *Zm00001eb078730*, and *Zm00001eb425600*, respectively; letters e and f: *Zm00001eb250900* and *Zm00001eb168350*; letters g and h: *Zm00001eb317090*, *Zm00001eb008880*; and letters i and j: *Zm00001eb354540* and *Zm00001eb167340*.

**Table 1 genes-15-01087-t001:** Information for the 39 *Z. mays* chitinase genes obtained from the Zm-B73-REFERENCE-NAM-5.0 genome.

Gene Name	Gene ID	Chr.	Gene Position	Transcript Length (bp)	Number of Amino Acids	GH Family	Isoelectric Point (pI)
*Chn1*	*Zm00001eb157820*	3	217635788–217636989	1202	295	18	4.06
*Chn7*	*Zm00001eb301500*	7	10810239–10811584	1346	312	18	7.11
*Chn12*	*Zm00001eb147120*	3	181373589–181374970	1382	311	18	9.12
*Chn13*	*Zm00001eb147140*	3	181400773–81402109	1269	299	18	5.92
*Chn14*	*Zm00001eb147150*	3	181403132–181404390	1155	294	18	4.93
*Chn15*	*Zm00001eb147160*	3	181489354–181490403	924	307	18	8.54
*Chn16*	*Zm00001eb301490*	7	10733581–10735052	1472	307	18	9.21
*Chn17*	*Zm00001eb047940*	1	246418756–246419832	1077	308	18	4.11
*Chn19*	*Zm00001eb420850*	10	110156286–110158162	1807	484	18	8.88
*Chn25*	*Zm00001eb283280*	6	139958834–139960420	1587	368	18	5.72
*Chn26*	*Zm00001eb283260*	6	139888026–139889202	1177	315	18	6.95
*Chn27*	*Zm00001eb167340*	4	7657537–7658988	1452	301	18	8.69
*Chn28*	*Zm00001eb250900*	5	205495288–205496493	1206	286	18	4.99
*Chn29*	*Zm00001eb168350*	4	13184755–13185856	1102	286	18	4.97
*Chn30*	*Zm00001d041426*	3	120119356–120120333	978	325	18	6.17
*Chn31*	*Zm00001eb358410*	8	148327731–148334525	1690	430	18	8.42
*Chn34*	*Zm00001eb169950*	4	22152266–22154411	1889	399	18	4.81
*Bk4*	*Zm00001eb317090*	7	139523498–139528852	1974	328	19	8.16
*Chn2*	*Zm00001eb272090*	6	92474902–92476065	1164	261	19	10.35
*Chn3*	*Zm00001eb228510*	5	65159105–65160362	1258	193	19	9.46
*Chn4*	*Zm00001eb340820*	8	42175395–42176288	807	268	19	8.09
*Chn5*	*Zm00001eb078720*	2	35057479–35058766	1160	278	19	7.86
*Chn8*	*Zm00001eb007850*	1	23078494–23079090	597	198	19	5.92
*Chn9*	*Zm00001eb270440*	6	80869472–80869762	291	96	19	9.1
*Chn10*	*Zm00001eb332350*	8	1145764–1146951	1188	285	19	6.97
*Chn11*	*Zm00001eb078740*	2	35087711–35095502	926	227	19	7.57
*Chn20*	*Zm00001eb228500*	5	65072200–65073359	1160	282	19	5.1
*Chn21*	*Zm00001eb346860*	8	90582152–90583599	1350	357	19	8.04
*Chn22*	*Zm00001eb002620*	1	7344153–7348877	1087	258	19	9.1
*Chn23*	*Zm00001eb354540*	8	132934929–132936181	1253	311	19	5.97
*Chn24*	*Zm00001eb022500*	1	86493293–86494456	940	210	19	4.79
*Cta1*	*Zm00001eb078730*	2	35084623–35085927	1202	280	19	8.44
*Ctb1*	*Zm00001eb425600*	10	129884568–129885888	1195	281	19	8.92
*EPR4*	*Zm00001eb246640*	5	186238039–186239256	1150	271	19	5.14
*Prp10*	*Zm00001eb272050*	6	92426037–92427305	1269	379	19	4.82
*Exo1*	*Zm00001eb266300*	6	42318401–42331903	2511	545	20	5.78
*Exo2*	*Zm00001eb008880*	1	27253118–27255816	2619	599	20	6.47
*Exo3*	*Zm00001eb288150*	6	158303978–158308928	2332	529	20	5.94
*Exo4*	*Zm00001eb365840*	8	170849713–170855399	2161	525	20	5.36

**Table 2 genes-15-01087-t002:** Chitinase gene-expression data obtained by RNA-seq and validated by qRT-PCR.

Gene ID	Synonyms	FC (Fold Change)	
		RNA-Seq	RT-qPCR
GH18			
*Zm00001eb157820*	Chitinase chem 1	4.92	1.64 ^a^
*Zm00001eb047940*	Chitinase 17	5.14	0.70 ^a^
*Zm00001eb167340*	Chitinase 27	13.41	3.61
*Zm00001eb250900*	Chitinase 28	4.17	3.76
*Zm00001eb168350*	Chitinase 29	2.71	3.49
*Zm00001eb169950*	Chitinase 34	2.57	1.81 ^a^
GH19			
*Zm00001eb317090*	Brittle stalk4	2.31	2.75
*Zm00001eb078740*	Chitinase 11	9.58	0.67 ^a^
*Zm00001eb272090*	Chitinase 2	5.16	2.35
*Zm00001eb346860*	Chitinase 21	7.27	2.51
*Zm00001eb354540*	Chitinase 23	6.69	4.99
*Zm00001eb078730*	Chitinase A1	4.44	2.03
*Zm00001eb425600*	Chitinase B1	4.11	4.02
GH20			
*Zm00001eb008880*	Exochitinase 2	4.14	3.25

The cyclin-dependent kinase gene (*Zm00001eb350890*) was used for normalization. Genes with a Fold Change (FC) value ≥ 2 in RT-qPCR analyses were considered as a validated gene. RT-qPCR values are presented as the mean of the two biological replicates used for RNA-seq analysis with three technical replicates per each biological replicate. ^a^ Gene expression was not validated.

## Data Availability

The original contributions presented in the study are included in the article/Appendix A, further inquiries can be directed to the corresponding author.

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
