# Peer review of "Genome-Wide Identification of a Maize Chitinase Gene Family and the Induction of Its Expression by Fusarium verticillioides (Sacc.) Nirenberg (1976) Infection"

_genes, 2024, doi:10.3390/genes15081087_

Round 1

Reviewer 1 Report

Comments and Suggestions for Authors

The manuscript titled "Genome-Wide Identification of A Maize Chitinase Gene Family and Its Induction of Expression by Fusarium verticillioides (Sacc.) Nirenberg (1976) Infection" contains valuable information on the characterization of genes coding for plant chitinases, poorly investigated enzymes, that have an immense role in plant combat against pathogen fungi. The study is well conducted and properly presented. However, I would suggest a couple of improvements in order to increase the overall quality of presentation.

Figure 1, Figure 2, and Figure 4: Too low resolution. Letters are not visible and pixelize with the magnification. Please increase the figure resolution.

Main title: I suggest inverting word order into "Genome-Wide Identification of a Maize Chitinase Gene Family and the Induction of Its Expression by Fusarium verticillioides (Sacc.) Nirenberg (1976) Infection"

L35: To the best of my knowledge, except for some algae (doi.org/10.1038/srep06162, DOI:10.1177/26.10.722047), chitin is not synthesized by plants. Therefore, the first sentence of Introduction should be reworded to follow the essence.

L45: The expression "Animal genomes containing chitins encode..." implies that genomes contain chitin, which is not true. Please amend the sentence to avoid conflicting information.

L53: Please avoid starting a paragraph with "These enzymes". Normally, any sentence can be started this way, but when starting a new paragraph, please provide full information.

L56-57: Are the authors implying here that chitinases of the remaining classes do not possess a chitin-biding domain? How do they act then? Please explain.

L197: Table title says that the table represents chitinase characteristics, but instead it shows genes coding for chitinases. Please rectify.

L198: Table 1: Please sort Chn genes according to their names in the following order: Chn1, Chn2, Chn3, and so on, instead of Chn1, Chn12...

L202: Please mention which proteins.

L206-209: The sentence structure is weak. Please rewrite to clarify.

L213-215: Figure caption suffers from poor syntax. Please reword.

L232, 233 and elsewhere in the Results: Please use past simple instead of present simple when describing the obtained results.

L349 and L494: Please capitalize "H" when starting a sentence with "β-hexosaminidases".

L482: Please extract the info outside the brackets. You don't need brackets here.

Comments on the Quality of English Language

Some sentences need rewriting to avoid confusion or to clarify their meaning. Past simple to be used in Results instead of present simple.

Author Response

Please see the atachment

Reviewer 2 Report

Comments and Suggestions for Authors

The manuscript by Cazares-Álvarez et al. mainly identified and predicted 39 maize chitinase genes characters through bioinformatic analyses. They further used qRT-PCR to analyze 14 maize chitinase genes expression patterns in response to F. verticillioides treatment. This study provides some clues that these maize chitinase genes may play a role in fungal infection. The following points should be addressed:

1.              The description of RNA-seq and qRT-PCR methods and results is not clear. In line 300, it mentions "In silico analysis of maize chitinase gene expression in 7-day-old F. verticillioides infected seedlings," but from the description below (lines 302-305), it seems that roots instead of seedlings were chosen for both RNA-seq and qRT-PCR analysis. However, in the Methods section, line 154, it appears that RNA samples used were from seedlings. If RNA-seq samples use whole seedlings and qRT-PCR samples use roots, the authors should explain and discuss the discrepancy.

2.              The source data mentioned in lines 302-303, "RNA-seq data from maize roots infected for 7 days with F. verticillioides (Fv) [40] were used to determine chitinase gene expression patterns in maize roots (Table S8)." However, Ref40 appears to be a doctoral dissertation, which is not available and cannot be reviewed for details.

3.              RNA-seq and qRT-PCR data should be presented with more details rather than just showing a fold-change value as seen in Table 2. Statistical analysis should be included.

Round 2

Reviewer 2 Report

Comments and Suggestions for Authors

My concerns have been addressed, and I have no further questions.